# Association of Tandem Repeat Number Variabilities in Subunit S of the Type I Restriction-Modification System with Macrolide Resistance in *Mycoplasma pneumoniae*

**DOI:** 10.3390/jcm11030715

**Published:** 2022-01-28

**Authors:** Joon-Kee Lee, Moon-Woo Seong, Ki-Wook Yun, Eun-Hwa Choi

**Affiliations:** 1Department of Pediatrics, Chungbuk National University Hospital, Cheongju 28644, Korea; leejoonkee@gmail.com; 2Department of Laboratory Medicine, Seoul National University Hospital, Seoul National University College of Medicine, Seoul 03080, Korea; MWSeong@snu.ac.kr; 3Department of Pediatrics, Seoul National University Hospital, Seoul National University College of Medicine, Seoul 03080, Korea; pedwilly@snu.ac.kr

**Keywords:** comparative genomics, epidemiology, macrolide resistance, *Mycoplasma pneumoniae*, whole genome analysis

## Abstract

*Mycoplasma pneumoniae* is one of the major pathogens responsible for pneumonia in children. Modern molecular genetics has advanced both the management and the epidemiologic study of this disease. Despite these advancements, macrolide resistance remains a global threat in the management of *M. pneumoniae* infection, for which the genetic background remains unrevealed. In this study, the result of whole genome analysis of 20 sequence type 3 (ST3) *M. pneumoniae* strains were examined to investigate the gene(s) associated with macrolide resistance. Overall, genetic similarities within *M. pneumoniae*, and especially ST3, were very high (over 99.99 %). Macrolide resistant ST3 strains shared 20 single nucleotide polymorphisms, of which one gene (*mpn085*) was found to be associated with resistance. BLAST comparison of *M. pneumoniae* revealed regular tandem repeat number variabilities between macrolide-susceptible and resistant strains for genes coding the Type I restriction-modification (R-M) system of subunit S (*HsdS*). Of the ten known *HsdS* genes, macrolide resistance was determined by the unique tandem repeat of *mpn085* and *mpn285*. In conclusion, the use of whole genome sequencing (WGS) to target macrolide resistance in *M. pneumoniae* indicates that the determinant of macrolide resistance is variabilities in the tandem repeat numbers of the type I R-M system in subunit S.

## 1. Introduction

*Mycoplasma pneumoniae* is one of the major pathogens responsible for pneumonia in children [1]. The role of antibiotics in the management of *M. pneumoniae* infection is controversial. However, the benefit of using appropriate antimicrobials for lower respiratory tract infections is generally accepted [2]. Macrolide has long been regarded the first line therapy for confirmed or expected *M. pneumoniae* pneumonia. However, macrolide resistance, which is known to be induced by a point mutation in the 23s rRNA component, has emerged as an on-going threat [3,4]. As the antibiotics levofloxacin and tetracycline have limited use in children due to possible side effects, this threat is a distinct challenge. Nevertheless, delayed antimicrobial treatment is associated with more severe and/or prolonged diseases [5]. Furthermore, there is increasing evidence of the associations between macrolide resistance and the development of *M. pneumoniae*-related extra-pulmonary diseases, as well as progression to prolonged and more serious lung disease [5,6,7].

Despite this situation, current knowledge concerning the mechanism of macrolide resistance is limited. Previous studies have proven an association between the 23s rRNA point mutation and macrolide resistance and the correlation has been verified by laboratory experiments [8]. Furthermore, molecular genetics have revealed that certain strains are associated with macrolide resistance [3,9,10]. Nevertheless, the specific genes or genetic changes that are responsible for macrolide resistance have not yet been revealed or proven. In this study, we investigated the whole genomes of 20 *M. pneumoniae* strains that are currently known to belong to sequence type 3 in respect of the existence of macrolide resistance. Discovering the genetic basis of macrolide resistance is likely to benefit the scientific community in terms of clinical treatment because of the ongoing increase in this form of resistance.

## 2. Materials and Methods

### 2.1. M. pneumoniae Strains

Twenty *M. pneumoniae* ST3 strains were selected from our previous work in which WGS was performed on 30 *M. pneumoniae* strains isolated from children with pneumonia in South Korea during the two epidemics from 2010 to 2016 (Table 1) [11]. As this study was targeted on the genetic differences that are associated with the existence of macrolide resistance, a single ST with a suitable combination of macrolide susceptible and resistant strains was selected. The gene sequences used were previously deposited in the National Center for Biotechnology Information (NCBI) database under the accession numbers.

### 2.2. Comparative Genomics-Phylogenetic Associations

To confirm the hypothesis that there would be genetic differences according to macrolide resistance, a phylogenetic tree was constructed prior to initiation of the study. Tree construction was attempted by several methods before CSI Phylogeny 1.4 was selected as having the best discrimination ability and applied with the following parameters: minimum depth at SNP positions (10x); minimum relative depth at SNP positions (10%); minimum distance between SNPs (10 bp); minimum SNP quality (30); minimum read mapping quality (25); minimum Z-score (1.96) [12]. CSI Phylogeny calls SNPs, filters the SNPs, carries out site validation, and infers a phylogeny based on the concatenated alignment of the high-quality SNPs. The output was visualized using Figtree 1.4.4 (http://tree.bio.ed.ac.uk/software/figtree/, accessed on 15 November 2021).

### 2.3. Comparative Genomics-Recombination/Reassortment

Recombination has been reported in previous literature, despite the extreme genetic stability of *M. pneumoniae*. However, the evidence for recombination driving the evolution of *M. pneumoniae* was considered insufficient [13]. Therefore, before proceeding for further analysis, we checked the possibility of recombination or reassortment. To obtain multiple sequence alignment for the full genetic sequences of the 20 strains, MAFFT version 7 was applied with default parameters [14]. The output was examined with RDP4 [15].

### 2.4. Single Nucleotide Polymorphism (SNP) and Insertion/Deletion (Indel) Analysis

To call SNPs and indels, completed genomes were first broken into 10-kb “reads” at 1-kb intervals and then aligned to the M129 reference (accession number NC000912) strain using BWA v0.7.7 [16]. Variant calling was performed using SAMtools [17]. The effects of the SNPs and indels in the resulting variant call format files were evaluated and annotated using SnpEff v3.3 [18].

The density of the SNPs in each strain was visualized by plotting the number of SNPs in a 1-kb interval. The SNPs were also manually scrutinized to obtain differences in the macrolide susceptible and resistant strains, as well as for the accordance of SNPs within identical macrolide resistant strains.

### 2.5. Comparative Genomics-Coding of DNA Sequences with Identical Functions

Based on the gene annotation of the Rapid Annotations using Subsystems Technology (RAST) and SEED, BLAST comparisons were applied to ascertain the differences between four macrolide susceptible strains (and the M129 reference strain, which is macrolide susceptible) [19,20]. Similarities between the strains were considered and the optimal cut-off for the similarity index was decided. After several comparisons, genes that commonly arose as possibly of interest were selected for further investigation. The similarities were also visualized by the SEED viewer as part of the comparison.

### 2.6. Gene Annotations

Although the functional comparisons from BLAST were based on RAST annotations, further manual annotations were considered. The NCBI Prokaryotic Genome Annotation Pipeline (PGAP) and the Kyoto Encyclopedia of Genes and Genomes (KEGG) database were used to obtain detailed annotations [21,22]. Furthermore, based on the genes that were flagged as significant and with respect to previous studies investigating these genes, all the possible genes that shared identical functions were included in the investigation [23,24,25]. The specific genes that are referred to throughout the manuscript are based on the original RefSeq locus tag for easy comparison with previous and further works [26].

### 2.7. Comparative Genomics-Sequence Alignments with Visualization

Multiple genes of interests were examined using the Pathosystems Resource Integration Center (PATRIC) which uses annotations based on RAST [27]. Categorized genes were aligned by the basic functions covered in PATRIC and alignments and phylogenetic associations were visualized with PATRIC.

### 2.8. Ethics Approval and Consent to Participate

The institutional review board of Seoul National University Hospital approved the study protocol for the prior work (IRB no. H-1012–007–341). Informed consent was exempted because nasopharyngeal aspirates were obtained as part of standard patient care. Only selected genomes from the *M. pneumoniae* that were collected previously were used in the current study.

## 3. Results

### 3.1. Phylogenetic Associations

The phylogenetic tree, which is based on twenty ST3 strains, revealed two large groups, although the distance estimates among the strains were relatively close (Figure 1). The two groups could be distinctly separated by macrolide resistance. The macrolide resistance group consists of 16 strains for which subgroups associated with similar years of collection could be assumed.

### 3.2. Evidence of Recombination and Genome Segment Reassortment

The output of the multiple sequence alignment for the twenty ST3 strains was evaluated for the possibility of recombination prior to further evaluation. No evidence of recombination by RDP4 was obtained (data not shown).

### 3.3. Variant Analysis with Plotting

The variant counts against the reference *M. pneumoniae* M129 ranged from 383 to 455 bp (median 423.5 bp) nucleotides (Table 2). The variant ranges of macrolide susceptible and resistant strains were 383–440 bps (median 392 bps) and 391–455 bps (median 426.5 bps), respectively. A comparison of the variant counts against the total length of the reference strain (816 394 bp) indicates overall similarities of approximately 99.95% between the strains.

In further evaluations that were based on annotations, variants categorized as low impact and modifier were put aside. These included synonymous, upstream, and stop retained variants. A total of 171 bps were shared universally between ST3 and the reference strains. When ignoring both the low impact/modifier variants and the common variants, the variant number variabilities among the ST3 strains ranged from 15 to 48 bps (median 32 bps). Therefore, the similarities among the ST3 strains in this study were calculated as >99.99%, with very few variants observed as compared to the whole length of the species *M. pneumoniae*.

To find hot spots with increased variabilities that may suggest differences according to macrolide resistance, the SNPs were plotted for each strain (Figure 2). The SNP count was defined as the number of SNPs in a 1-kb interval. On the whole, a few common but irregular SNP patterns were noticed. Nevertheless, differences between macrolide susceptible and resistant strains were not clearly visualized, suggesting the need for further in-depth investigation.

### 3.4. Variant Analysis between Macrolide Susceptible and Resistant Strains

As we hypothesized that variation should be shared according to macrolide resistance, the variants were manually compared. No variation was found to be shared universally among the susceptible strains and the reference strain. Of the macrolide resistant strains, 20 common variations were not shared with either the reference strain or the macrolide susceptible strains (Appendix A). Of these, missense variation was the most common (16, 80.0%) with conservative inframe insertion/deletion, frameshift, and stop gained variations also observed (one each at 5%). Except for the inframe insertion/deletion variations, the substitutions in the 18 other variants were identical. The translated functions of the majority of the genes were hypothetical based on gene annotations (12, 60.0%).

### 3.5. Comparisons of Genes with Identical Annotation

Considering the extremely high similarities and lack of evidence for recombination or rearrangement in the strain studies, it is reasonable to assume that the coding genes are aligned in the same order and in almost the same bp positions. BLAST was used to compare each gene in the macrolide resistant strains against those in the macrolide susceptible strain, and a few genes were found to show similarities below 99.0%. After repetitive and sequential comparisons, two coding genes were indicated as the core difference between the macrolide susceptible and resistant strains, *mpn089* and *mpn285* (Appendix A), for which the functional annotation was Type I restriction-modification (R-M) system, specificity subunit S (*HsdS*).

### 3.6. Comparisons of Genes Associated with the Type I Restriction-Modification System, Subunit S

We subsequently changed focus to investigate the specific differences in the type I R-M system. First of all, multiple sequence alignment was attempted for the two genes, after which the nucleotides were translated into amino acid sequences for visualization of the alignments.

The length (nucleotide) of *mpn089* ranges from 1092 to 1140 bps (reference strain 1140 bps, location 111,478–112,617 bps). Without any other variations, the difference in length was explained solely by inframe deletions of the “ELSA” tandem repeats in the amino acid sequences (Figure 3A). Based on the reference sequence, the number of deletions in the tandem repeat ranged from 0 to 4. Significant differences were observed when the deletion variabilities were compared with the presence of macrolide resistance, with the deletion numbers ranging from 0 to 1 in the macrolide susceptible strains and from 2 to 4 in the macrolide resistant strains.

The length of *mpn285* ranges from 1254 to 1446 bps (reference strain 1290 bps, location 340,244–341,533 bps). Roughly, the total length of the macrolide susceptible strains (ranging from 1254 to 1374 bps) was shorter than that of the macrolide resistant strains (commonly 1446 bps). The results show that macrolide susceptible strains, including the M129 reference strain, had deletions of the “TELS” tandem repeats that are not observed in the macrolide resistant strains (Figure 3B). The number of deletions ranged from six (strain 11-473) to 16 (strains 11-107 and 11-994). Interestingly, a few threonine to alanine changes that originated from adenine to guanine SNP changes were scattered around the regions with tandem repeats (Appendix A).

*M. pneumoniae* is known to include 10 *HsdS* genes. Therefore, the eight other *HsdS* genes were also examined. The two genes *mpn289* and *mpn615* also showed variation in the tandem repeats (Appendix A). Both of the translated amino acid products shared the tandem repeat “ELSA,” which is identical to that seen in MPN089. However, a clear distinction of macrolide resistance could not be associated with the number of tandem repeats.

### 3.7. Type I Restriction-Modification System, Subunits R and M

We then investigated the other specificity subunits R (*HsdR*) and M (*HsdM*) that comprise the I R-M type system. Exploited genes that were found to be in association (including putative) with *HsdR* and *HsdM* were: *HsdR* (*mpn109*, *mpn110*, *mpn345*, *mpn346*, and *mpn347*) and *HsdM* (*mpn107*, *mpn108*, *mpn111*, and *mpn342*). Except for *mpn109*, all other examined genes were identical in both the ST3 strains and the M129 reference strain. The phylogenetic association of the sequence alignment for *mpn109* indicated more similarities in the three strains as compared to other strains (data not shown): 10-1257 (S), 11-212 (R), and 12-060 (R). However, associations with macrolide resistance were not observed for all genes associated with *HsdR* and *HsdM*.

## 4. Discussion

The whole genome analysis of certain types of *M. pneumoniae* revealed that the tandem repeat number changes in the genes that code the type I restriction-modification system, and in particular, subunit S is associated with macrolide resistance. We believe that our findings will have implications for furthering our understanding of the underlying genetic basis of resistance. Such an understanding will help to answer ongoing questions as well as hopefully support the practical management of *M. pneumoniae* infections in children and adults.

As *M. pneumoniae* is one of the most difficult bacterium to culture, molecular genetics has had an enormous impact on both the management of the disease and epidemiologic study [28]. The diagnosis of *M. pneumoniae* infection is currently generally based on nucleic acid amplification tests [1]. Additionally, despite controversies concerning the impact of macrolide resistance, the presence of a point mutation in 23s rRNA is considered in the management (especially the antibiotic selection) of *M. pneumoniae* pneumonia in children and adolescents [2].

Apart from diagnosis and treatment, methods that use molecular genetics, such as P1 typing, multiple-locus variable-number of tandem-repeats analysis (MLVA), and multilocus sequence typing (MLST), are applied in modern epidemiologic studies [29]. MLVA and MLST have increasing discrimination power compared to P1 typing. Studies that combine all three molecular genetic techniques generally produce results that are in concordance with each other. Recent studies have demonstrated that certain serotypes that were obtained using MLST are associated with macrolide resistance, with both ST3 and ST17 associated with high macrolide resistance in Asian countries [3,4,30]. On the other hand, the strain ST33, which has markedly decreased macrolide resistance, appears to be increasing in Japan [31]. Improvements in molecular genetics have made it possible to apply WGS to the species *M. pneumoniae*, with studies verifying certain genetic differences in the traditionally categorized P1 types 1 and 2. Furthermore, advances in the discrimination ability have allowed phylogenetic associations between global strains to be visualized [11,13,25]. However, limited attention may have been paid to *M. pneumoniae* because of its known stability, with almost no recombination or rearrangements.

The underlying genetic basis of macrolide resistance has not been elucidated beyond the known point mutation in the 23s rRNA component. Therefore, our attention was drawn to using WGS in an attempt to understand the genetic basis of macrolide resistance in *M. pneumoniae* because of the scarceness of the literature and the clinical significance of macrolide resistance in real-world practice. Based on the known ST, we hypothesized that studying the same ST types with different levels of macrolide resistance by WGS would reveal the association between the differences in certain genes and macrolide resistance. The type I R-M system turned out to be the anticipated gene of interest.

Living organisms need to defend against foreign DNA, such as that borne by bacteriophages. This is provided by the R-M system in bacteria and other prokaryotic organisms [32]. *M. pneumoniae* is known to possess a type I R-M system with putative sequences that are related to the type II R-M system [23]. Type I R-M systems are comprised of three polypeptides that are associated with restriction (R), modification (M), and specificity (S) [32], and type I R-M enzymes are pentameric proteins with the composition (2R + 2M + S). The subunit R cleavages the recognition site and subunit M methylates the designated site by adding appropriate nucleotides. However, the sites at which these processes can occur first need to be recognized and subunit S plays the role of identifying such regions.

The individual target recognition domains (TRDs) of S subunits can be shuffled into different combinations by genetic rearrangement, which means that the number of microbes in close proximity can potentially affect mutation far more than the number of *HsdS* genes [32]. Specificity changes can be achieved in several ways: TRD exchanges, gap length changes, homodimeric S subunits, the circular permutation of S subunits, and recognition sequence orientations. The findings of this study indicate that the specificity differences in *M. pneumoniae* originate from changes in the gap lengths. Price et al. showed that the repeat of certain base-pair sequences actually does change specificity in terms of both restriction and modification [33]. By modifying the number of tandem repeats of an *HsdS* from *Escherichia coli*, the researchers sufficiently explained the differences in sequence recognition. The extra four amino acids in the middle of the EcoRl24/3 *HsdS* gene product, which in a cc-helical configuration would extend to 0.6 nm, were sufficient to explain the differences in sequence recognition. Since the R-M system is the pertinent component in defending DNA from mutations and the mechanism of macrolide resistance in *M. pneumoniae* is a point mutation in the 23s rRNA component, it is plausible that the tandem repeat variabilities in the gene associated subunit S seems to be the main key to explaining macrolide resistance [34]. However, we are not sure whether the differences in the tandem repeat numbers are innate or acquired. Antibiotic pressure could have played a role or certain tandem repeat patterns may have been coupled with the existence of macrolide resistance [3]. Multiple passages could be attempted with *M. pneumoniae* to ascertain whether the number of tandem repeats could be changed. Furthermore, manipulating the number of tandem repeats via bioengineering could be attempted to see the result of the changes [25].

The results of our study answer a few questions raised by previous studies regarding molecular genetics. Xiao et al. also studied the tandem repeats of *HsdS* in their previous work [25], and found a 12-bp tandem repeat within the dimerization domains of seven *HsdS* genes. The tandem repeat copy numbers were found to vary in six of the *HsdS* genes in all strains and in different passages for the same strain. However, they were not able to expand their results in association with macrolide resistance, despite suggesting the importance of epigenetics in this species. Their results partially prove that the performance of multiple passages can alter the number of tandem repeats. In addition, having reviewed the tandem repeat numbers in their strains from the perspective of macrolide resistance, our findings were completely consistent. Another interesting fact is that the *mpn089* gene, which is one of the genes of interest in this study, has previously received attention under molecular genomics. Five loci were considered when the MLVA method was initially proposed, whereas only four loci are currently considered for classification [35,36]. The locus *mpn1* was later removed from the MLVA scheme because of the large number of variabilities (and the decreasing discrimination power) [37,38]. The unincluded *mpn1* gene, *mpn089,* is the focus of this study. The results of our study suggest that the tandem repeat number variability of *mpn1* (=*mpn089*) may be associated with macrolide resistance. Therefore, the findings of this study may partly explain some of the inaccuracies that were observed during the original MLVA scheme, in addition to the questions surrounding the tandem repeat numbers of *HsdS*.

The practical applications of our findings should also be considered. As our findings are limited to the molecular genetics of *M. pneumoniae*, practical applications may be limited. Furthermore, as the strains investigated in this study originated from children or adolescents, a question arises concerning whether the results of this study can also be extended to adults. Currently, we believe that our findings may provide a partial clue to a previously unanswered questions, such as: Why does the rate of macrolide resistance in *M. pneumoniae* go up and down? Some previous research has suggested antibiotic pressure as an explanation [1]. Furthermore, a clonal expansion of certain strains has also been suggested [3]. Nevertheless, there have been no convincing answers to this pertinent question. However, with the knowledge of the underlying genetic mechanism of macrolide resistance, future studies may be able to address this issue in more detail. By demonstrating the underlying genetic variability, the role of genetic variability in stabilizing pathogen structure and pathogenesis can be understood further. In addition, future development of safe and effective antibiotics for children against macrolide-resistant *M. pneumoniae* may aid from our findings. Estimates of *M. pneumoniae* infection range from 2% to 12% among adults with community-acquired pneumonia, which implies the importance of this specific pathogen even in adults [39,40]. Even though our results may be of limited applicability to adults, we believe that our findings will aid in the management of adult *M. pneumoniae* pneumonia infections to some extent [41].

Our study is ultimately limited by the fact that we have been able to investigate only the ST3 strains. The findings of this study may not be consistent in other strains, especially the P1 type 2 strains, because ST3 is a P1 type 1 strain. The macrolide resistance of P1 type 2 strains is relatively low compared to that of P1 type 1 strains, although resistance has increased recently in certain regions [42,43,44]. Therefore, other mechanisms of macrolide resistance, including but not limited to the R-M system, may occur in P1 type 2 strains. Secondly, the strains are from a limited region, South Korea. Strains from different areas should therefore be studied to examine whether the findings of this study are universal throughout the species *M. pneumoniae*.

## 5. Conclusions

Through WGS of *M. pneumoniae*, we determined that macrolide resistance is associated with variability in the tandem repeat numbers of subunit S in the type I R-M system. The genome-wide structural basis of macrolide resistance is likely to contribute to the better understanding of pathogenesis and to the advancement of treatment strategies for *M. pneumoniae* infections.

## Figures and Tables

**Figure 1 jcm-11-00715-f001:**
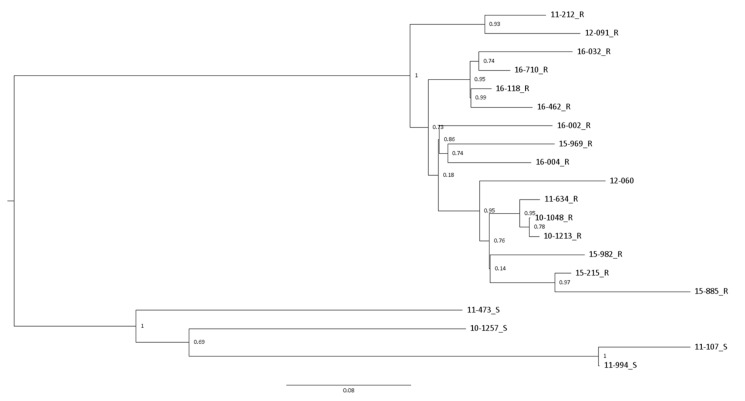
Phylogenetic tree generated from four macrolide susceptible and 16 macrolide resistant *M. pneumoniae* ST3 strains using CSI Phylogeny. Each are labelled in the order of year of acquisition, individual strain number, and either S or R, designating macrolide susceptible or resistant, respectively. Bootstrap values are shown. ST, sequence type; S, macrolide susceptible; R, macrolide resistant.

**Figure 2 jcm-11-00715-f002:**
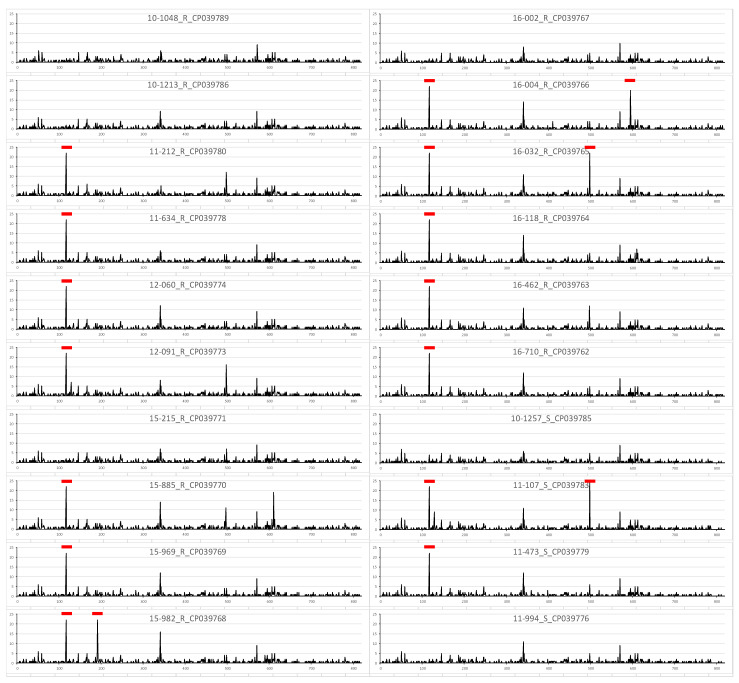
Density of SNPs in four macrolide-susceptible and 16 macrolide-resistant *M. pneumoniae* ST3 strains against the reference genome of *M. pneumoniae* M129. Each gene is labeled in order of the year of the acquisition, individual strain number, the presence of macrolide resistance, and the accession number. The x-axis represents the physical distance along each strain, split into 1-kb windows. The y-axis indicates the number of SNPs. The regions with relatively high density are labeled with red bars. SNP, single nucleotide polymorphism; ST, sequence type; S, macrolide susceptible; R, macrolide resistant.

**Figure 3 jcm-11-00715-f003:**
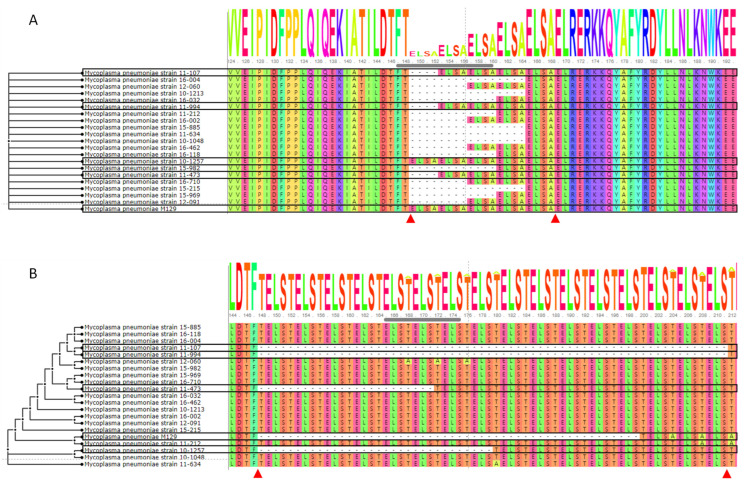
Alignment of amino acid sequences with different tandem repeat numbers between macrolide susceptible and resistant strains. (**A**) Amino acid sequence alignment of MPN085 and (**B**) MPN285. Macrolide susceptible strains and corresponding amino acids are boxed in black. The start and end positions of tandem repeats of interest are highlighted with arrow heads (red).

**Table 1 jcm-11-00715-t001:** Characteristics of ST3 strains including macrolide resistance, total length, and NCBI accession numbers.

Strain	Year Collected	Macrolide Resistance	Total Length (bps)	NCBI Accession
10-1048	2010	Resistant	816 465	CP039789
10-1213	2010	Resistant	816 521	CP039786
10-1257	2010	Susceptible	816 333	CP039785
11-107	2011	Susceptible	816 346	CP039783
11-212	2011	Resistant	816 503	CP039780
11-473	2011	Susceptible	816 518	CP039779
11-634	2011	Resistant	816 551	CP039778
11-994	2011	Susceptible	816 304	CP039776
12-060	2012	Resistant	816 506	CP039774
12-091	2012	Resistant	816 510	CP039773
15-215	2015	Resistant	816 388	CP039771
15-885	2015	Resistant	816 420	CP039770
15-969	2015	Resistant	816 389	CP039769
15-982	2015	Resistant	816 495	CP039768
16-002	2016	Resistant	816 530	CP039767
16-004	2016	Resistant	816 561	CP039766
16-032	2016	Resistant	816 471	CP039765
16-118	2016	Resistant	816 467	CP039764
16-462	2016	Resistant	816 525	CP039763
16-710	2016	Resistant	816 537	CP039762

NCBI, National Center for Biotechnology Information; ST, sequence type.

**Table 2 jcm-11-00715-t002:** SNPs and indel analysis of ST3 strains against the reference M129 *M. pneumoniae*.

Strain	Upstream	Synonymous	Missense	Start/Stop	Inframe	Frameshift	Total
10-1048_R	89	105	153	13	6	25	391
10-1213_R	93	102	154	16	7	25	397
10-1257_S	92	95	151	15	5	25	383
11-107_S	114	107	172	15	9	23	440
11-212_R	118	108	154	13	7	25	425
11-473_S	116	97	141	15	5	25	399
11-634_R	110	103	154	16	6	25	414
11-994_S	92	99	151	12	7	24	385
12-060_R	119	104	160	15	7	25	430
12-091_R	130	104	162	16	7	27	446
15-215_R	95	106	155	13	7	27	403
15-885_R	130	108	170	15	7	25	455
15-969_R	114	104	157	14	8	25	422
15-982_R	142	108	157	14	8	25	454
16-002_R	92	104	156	12	8	25	397
16-004_R	116	114	163	14	8	27	442
16-032_R	121	106	166	17	6	25	441
16-118_R	126	100	156	14	7	25	428
16-462_R	128	101	159	14	7	25	434
16-710_R	115	100	158	14	7	25	419

SNP, single nucleotide polymorphism; ST sequence type; S, macrolide susceptible; R, macrolide resistant.

## Data Availability

The data presented in this study are available on reasonable request from the corresponding author.

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
