# Peer review of "Association of Tandem Repeat Number Variabilities in Subunit S of the Type I Restriction-Modification System with Macrolide Resistance in Mycoplasma pneumoniae"

_jcm, 2022, doi:10.3390/jcm11030715_

Round 1

Reviewer 1 Report

The manuscript „Association of tandem repeat number variabilities in subunit S of the Type I restriction-modification system with macrolide resistance in Mycoplasma pneumoniae” is well written. Mycoplasma pneumoniae is one of the major human pathogens and modern molecular genetics gives a lot of information about the pathogen, especially about antibiotic resistance. Macrolide resistance is a global threat and this manuscript gives an insight into its examination. One minor change is to rewrite the sentence in line 237, where word “methods” is used twice.

Reviewer 2 Report

INTRODUCTION

- The authors could also highlight the relevance of MP in extra-respiratory diseases, especially in children, as reviewed in “Extra-pulmonary diseases related to Mycoplasma pneumoniae in children: recent insights into the pathogenesis.” (Curr Opin Rheumatol. 2018 Jul;30(4):380-387).

- Indeed, this aspect is important for the main topic of this study, since some of the extra-pulmonary disorders/complications of MP infections may be in part related to antibiotic (macrolide) resistance (Medicine (Baltimore). 2021 Mar 19;100(11):e24894) or delayed (J Microbiol Immunol Infect. 2020 Feb;53(1):188-189. doi: 10.1016/j.jmii.2019.04.011., highlighting this aspect based on the findings from the following study: J Microbiol Immunol Infect . 2019 Apr;52(2):329-335.) treatment as recently discussed by some authors.

METHODS

- the study approach is clearly described. The ethical aspects are also clarified.

RESULTS

- the results are well presented in detail.

DISCUSSION

- Conversely, I think that the discussion is somehow dispersive. Therefore, I would invite the authors to reorganize it extensively, also taking in account the main clinical readership of this journal. Indeed, whereas the methods and results are technically presented, as it must be, in this case the discussion should also provide the opportunity to explain the main findings in more general terms and to suggest the potential practical (and clinical) impact of the present analysis.

- Following the previous comment, I would suggest the authors to start the discussion by clearly listing and highlighting their main findings. Then, they may discuss these (bullet) points one by one in light of the available literature.

- a separated conclusion section would be advisable in my opinion.

- In the method the authors state “Twenty M. pneumoniae ST3 strains were selected from our previous work in which 53 WGS was performed on 30 M. pneumoniae strains isolated from CHILDREN with pneumonia 54 in South Korea during the two epidemics from 2010 to 2016 (Table 1) [8].” I think the authors should discuss if these results could be applied to the pediatric population only or how these can be extended to adults.

- As mentioned above, the authors should discuss more the potential practical applications of their findings, if any.

- Some very recent papers provided additional contribution by MLVA on the topic starting from pediatric samples (e.g. J Glob Antimicrob Resist . 2022 Jan 8;S2213-7165(22)00001-7.; Biomed Environ Sci. 2020 Dec 20;33(12):916-924; etc.) : I think these articles may be useful to improve the discussion and may help the authors to highlight the novelty of their results.

Round 2

Reviewer 2 Report

I have no additional comments.